# Prevalence, trend and associated factors of obesity-related cancers among U.S. adults with metabolic syndrome: Evidence from the National Health and Nutrition Examination Survey 2001–2018

Harun Mazumder[1,2], Maidul Husain[3], Md. Faruk Hossain[1], Sultan Mahmud[4]*

1 Institute of Statistical Research and Training, University of Dhaka, Dhaka, Bangladesh, 2 School of Public Health, Louisiana State University Health Sciences Center-New Orleans, New Orleans, Louisiana, United States of America, 3 Bangabandhu Sheikh Mujibur Rahman Science and Technology University, Gopalganj, Bangladesh, 4 International Centre for Diarrhoeal Disease Research, Dhaka, Bangladesh

* smahmud@isrt.ac.bd

## Abstract

### Introduction

This study evaluated the prevalence, associated factors and trends in the prevalence of obesity-related cancer (ORC) among U.S. adults with metabolic syndrome (MetS) and age ≥20 years.

### Methods

This study used cross-sectional data from the 2001–2018 National Health and Nutrition Examination Survey. The total period analyses included prevalence estimation, chi-square tests for comparing ORC vs non-ORC within subgroups, and a multivariable-logistic regression model to evaluate associated factors of ORC. For trend analysis, the total period was divided into three time periods: 2001–2006, 2007–2012 and 2013–2018. Age-standardized prevalence of ORC in each time period was calculated.

### Results

The ORC prevalence was 35.8% representing 4463614 adults with MetS. A higher odds of ORC was observed among females (OR = 7.1, 95% CI = 4.9–10.3) vs males, Hispanic (OR = 2.9, 95% CI = 1.7–4.8) and non-Hispanic Black (OR = 2.7, 95% CI = 1.8–4) vs non-Hispanic White, age ≥60 (OR = 5.4, 95% CI = 1.9–15.4) vs age 20–39 years. Individual ORCs were thyroid (10.95%), breast (10%), uterine (9.18%), colorectal (7.86%), ovarian (5.74%), and stomach (0.80%). The age-standardized prevalence of ORC was observed stable in three time periods (30.6%, 30.3% and 30.7%). However, an increasing trend was seen for thyroid, uterine, colorectal and ovarian cancers while decreasing trend for breast cancer. Hispanic people showed a significant increasing trend of ORC (p = 0.004).

**Data Availability Statement:** Data sets are publicly available in Centers for Disease Control and Prevention website (https://wwwn.cdc.gov/nchs/nhanes/).

**Funding:** The author(s) received no specific funding for this work.

**Competing interests:** The authors have declared that no competing interests exist.

## Conclusions

ORC was found significantly higher among female, Hispanic, non-Hispanic black and older people with MetS. The stable temporal trend of overall ORC, with an increasing trend in certain ORCs, makes the disease spectrum a public health priority. The findings imply the importance of intensifying efforts to reduce the burden of MetS comorbidities among U.S. adults.

## Introduction

Obesity and cancer are two interlinked major public health issues in the United States and globally [1, 2]. Overall cancer is the second leading cause of death in the United States [3], and nearly 40% of all cancer diagnoses in the U.S. are obesity-related cancer (ORC) [4]. Excess body fat can lead to ORC by causing changes in the body such as chronic inflammation and hormonal imbalance, and altering microbiome composition [5]. Obesity has been found to be linked to a higher risk for cancer in at least 13 anatomic sites [2, 4, 6, 7]. Research shows that ORCs have biological properties that are associated with poor prognosis [8, 9]. The average incremental cost of treating ORC is 2.1 times higher than that of non-ORC [10]. A substantial burden of ORC morbidity and mortality is attributable to poor metabolic health [11]. Metabolic Syndrome (MetS) is the current indicator of metabolic health [12, 13].

MetS is referred to as a group of conditions that includes three or more of the following conditions: diabetes, hypertension, abdominal obesity, hypertriglyceridemia, or low levels of high-density lipoprotein (HDL) [12]. MetS is very common and about one-third of U.S. adults have it [14]. It can independently act as the major risk factor for cardiovascular disease (CVD) and type 2 diabetes in adults [15, 16]. CVD is the most common cause of death among cancer survivors [17], which can be attributed to MetS, a long-term complication of curative cancer treatment [18]. An increasing number of studies suggest that MetS is associated with ORC [19–22]. Obesity or overweight and physical inactivity are major risk factors for MetS [23]. Although being overweight or obese does not mean one will definitely develop ORC, the longer duration of being overweight increase the risk [4, 24]. Some modifiable factors such as maintaining a balanced diet and engaging in regular physical activity can positively influence the morbidity of MetS [25].

Research shows a slightly increasing trend in MetS prevalence among U.S. adults during 2011–2016 [14], which may be due to an increasingly sedentary lifestyle [26]. Although overall cancer incidence has declined since the 1990s, an increase in the incidence rate of ORCs, excluding colorectal cancer, was observed among U.S. adults of age 20–74 years in 32 states during 2005–2014 [4]. The advances in knowledge of ORC etiology and improvement in medical treatment to control or prevent serious comorbidities associated with MetS should reduce ORC prevalence over time. However, less is known about the temporal trend in the prevalence of ORC among adults with MetS, though MetS is steadily increasing among U.S. adults. Furthermore, there is a link between obesity in adult cancer patients and several challenges including a reduced likelihood of obtaining cancer screening, difficulties identifying cancer due to overlapping fat tissue, and receiving inadequate chemotherapy doses [8]. Knowing the current prevalence, related risk factors and temporal trends of ORC may identify high-risk groups with MetS who would benefit from adherence to recommended cancer screening guidelines and minimize the risk for ORC. This study aimed to determine the prevalence and associated factors of ORC, relative to non-ORC, based on recent data, and further evaluated the temporal

trends in ORC among U.S. adults with MetS. To better inform future research on MetS and ORC, it is also important to know the prevalence and trend of currently existing individual ORCs (e.g., colorectal cancer) among U.S. adults with MetS. Therefore, the prevalence and trends of individual ORCs were also evaluated.

## Methods

### Study population

Data were collected from National Health and Nutrition Examination Survey (NHANES) 2001–2018. NHANES uses a complex, multistage, probability sampling design to collect a representative sample of the non-institutionalized U.S. population. NHANES collects data in two-year cycles. More information on NHANES design, questionnaires, and examination procedures is provided elsewhere [27]. The current study was not reviewed by the Institutional Review Board since the data analyzed are de-identified and publicly accessible. However, the NHANES protocol was reviewed and approved by the National Center for Health Statistics (NCHS) Ethics Review Board [28]. Adults of age 20 and over with diagnoses of MetS and cancer were included in this study. Pregnant women were excluded from the study as they tend to have temporary MetS. Data from the 9 NHANES cycles were combined to obtain a total period 2001–2018. For evaluating the temporal trend in ORC prevalence, we divided the total period into three time periods of equal length: 2001–2006, 2007–2012 and 2013–2018.

The individuals with cancer diagnoses were identified who answered "yes" to the question "Have you ever been told by a doctor or health professional that you had cancer or malignancy of any kind?" The MetS was defined based on the National Cholesterol Education Program Adult Treatment Panel-III guidelines [12]. A diagnosis of MetS was determined when three or more of the five conditions were present: 1) elevated waist circumference ($\geq$88 cm for women and $\geq$102 cm for men), 2) taking medication to reduce triglyceride levels or having elevated triglycerides ($\geq$150 mg/dL), 3) low HDL cholesterol (<50 mg/dL for women and <40 mg/ dL for men) or taking HDL boosting medication, 4) elevated blood pressure (systolic $\geq$130 mmHg, or diastolic $\geq$85 mmHg, or both) or taking antihypertensive medication, 5) elevated fasting glucose ($\geq$100 mg/dL) or drug treatment for elevated glucose. Note that participants that were selected to give a fasting blood sample constructed the smallest survey subsample, and appropriate probability sampling weights were calculated by NHANES to make it representative of the U.S. population [29]. Participants who were unable to obtain fasting glucose levels had missing values in sampling weights and were excluded from this study.

### Study variables

The response variable was the ORC status (ORC, non-ORC). A diagnosis of brain, bladder, esophagus, kidney, endometrial, thyroid, ovarian, breast, liver, gallbladder, stomach, colorectal, or pancreatic cancer was required for the presence of an ORC to be established [2, 4, 6, 7]. All other cancers were considered non-ORC. Participants could list up to three separate cancer diagnoses when asked if they had cancer. The respondent was labeled as having an ORC even if the other two cancers were not obesity-related. The associated factors included: age (20–39, 40–59 and 60+), gender (male, female), race (Hispanic, non-Hispanic white, non-Hispanic black, and Other), education (high school graduate or less, some college degree, some college or above), annual household income ($35,000, $35,000 to $74,999, or $75,000+), country of birth (US-born, Mexico-born, and others), insurance status (yes, no), physical activity (yes: moderate or vigorous activity, no: otherwise), smoking (never, former, current), and alcohol use (never, former-drinker, mild, and heavy-drinker). The 'Other' races were excluded due to the limited sample size. Participants who did not smoke at least 100 cigarettes in their lifetime

were considered never smokers. The current and former smokers were those who smoked at least 100 cigarettes and reported their current smoking status as yes and no, respectively. Participants who did not drink at least 12 drinks in their lifetime were never drinkers; Former drinkers were those who had 12 drinks in their lifetime but did not have 12 drinks in the last year; heavy were those who had 12 drinks in the last year with a frequency of drinks ≥5 in at least one day; mild drinkers had this frequency <5.

## Statistical analysis

All analyses in this study were adjusted by appropriate sampling weights (fasting sampling weights) to ensure nationally representative estimates [29]. The analyses were conducted in R using survey package [30]. The prevalence of ORC for sociodemographic and behavioral characteristics was calculated. The Rao-Scott chi-square tests were performed to determine differences in these characteristics between ORC and non-ORC for categorical factors and the t-test for a continuous variable. To determine the related factors of ORC, we performed univariate logistic regressions and a multivariable logistic regression model. The age-standardized prevalence of ORC, overall and within subgroups, was calculated in each of the three time periods by applying direct method of standardization based on the U.S. 2000 population [31]. Age standardization was applied to adjust for differences in population age distributions across the time periods. Hence, the prevalence estimates were comparable across the three time periods. The differences in the age-standardized prevalence between time periods were also calculated with corresponding 95% confidence intervals using the two-proportion Z-test. The linear trends in ORC prevalence were assessed using univariate logistic regressions after regressing 'ORC status' on the continuous variable 'time period' (e.g., 1, 2, 3). Age standardization was applied before fitting these logistic regressions. Linear trends were assessed on the logit scales. More specifically, the P-values for linear trends were calculated considering a one-sided t-test based on positive (increasing trend) or negative (decreasing trend) values of the rate of change in log-odds from the logistic regressions.

## Result

### Study participants and prevalence of ORC

In the total study period 2001–2018, a total of 91351 participants were screened for eligibility. After removing incomplete data (who were not selected to give a fasting blood sample), 30065 participants were found to be eligible initially. Next, 19536 participants were excluded due to being pregnant women, having metabolic syndrome, and being aged < 20 years. Further, a total of 9146 participants with no history of cancer were removed to obtain a final analytical sample of 1383 participants (Fig 1). The analytical sample sizes in the three time periods (2001–2006, 2007–2012, and 2013–2018) were 354, 507, and 522, respectively (Fig 1).

Table 1 shows the number of participants and the prevalence of ORC, overall and within each subgroup, during the total study period 2001–2018. A lower prevalence of ORCs (n = 544, 35.8%) was observed compared to non-ORCs (n = 839, 64.2%) representing 4463614 and 8021267 U.S. adult populations with MetS, respectively. The mean age of the adults with and without ORC was 67.40 years and 64.43 years, respectively. Among 720 females with MetS, the prevalence of ORC was 52.3% (n = 412). Among total adults with MetS within each subgroup characteristic, the higher prevalence of ORC also existed in: Hispanics (n = 106, 53.34%), Non-Hispanic Blacks (n = 94, 52.5%), Mexican-born (55.5%), and never-drinkers (54.35%). The significant differences (p<0.05) between the prevalence of ORC and non-ORC were observed according to gender, race, age, education, income, country of birth, physical activity and alcohol use.

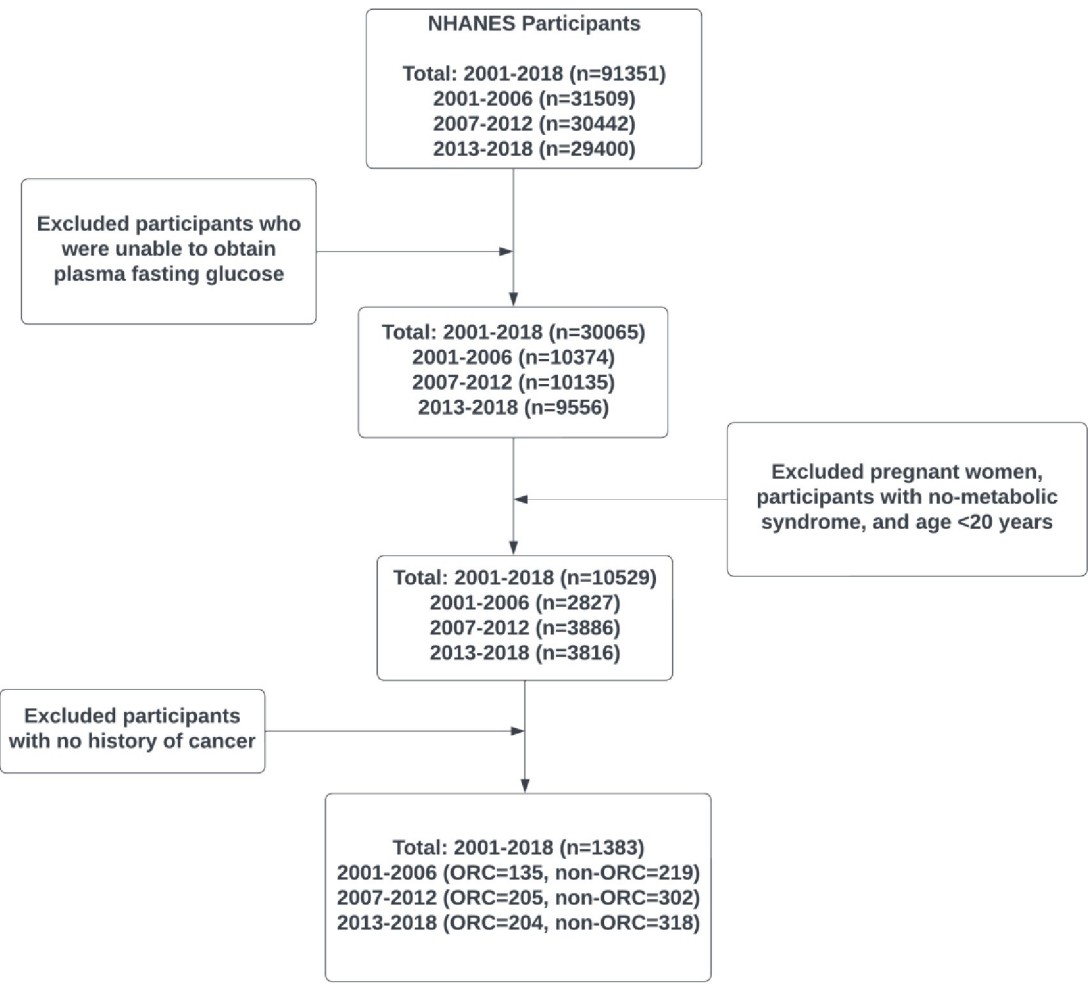

**Fig 1. Time period construction diagram and selection of analytical sample.**

## Associated factors of ORC among adults with MetS

Multivariable logistic regression was used to analyze the associated factors of ORC among U.S. adults with MetS during 2001–2018 (Table 2). Among the seven factors, three (Gender, Race, and Age) were found to be significantly associated with ORC. The odds of having ORC in females was approximately 7.1 times higher than in males (95% CI = 4.9–10.3; p<0.001). For race, the odds of having ORC among Hispanic and non-Hispanic black were 2.9 times (95% CI = 1.7–4.8; p<0.001) and 2.7 times (95% CI = 1.8–4; p<0.001) higher than the non-Hispanic. The adults of age 60 years and over were 5.4 times (95% CI = 1.9–15.4; p<0.001) more likely to have ORC compared to adults aged 20–39 years. Furthermore, a higher risk of ORC, though not statistically significant, was seen in time period-3 participants compared to period-1, age 40–59 vs age 20–39, those who were highly educated, and who had lower income. Those who had health insurance, and who were physically active had less risk for ORC, though not statistically significant.

## Temporal trends in the age-adjusted prevalence

The overall age-adjusted ORC prevalence in all three periods remained stable (Table 3). But the age-standardized ORC prevalence increased gradually among females, increasing from

**Table 1. Demographic and behavioral factors associated with ORC status among U.S. adults with metabolic syndrome, NHANES 2001–2018.**

| Characteristics | N[a] | ORC | non-ORC | P-value[c] |
|---|---|---|---|---|
| | | N (wt %)[b] | N (wt %)[b] | |
| Projected population[d] | | 4463614 | 8021267 | <0.001 |
| Total | 1383 | 544 (35.8) | 839 (64.2) | |
| **Gender** | | | | |
| Male | 663 | 132 (14.92) | 531 (85.08) | <0.001 |
| Female | 720 | 412 (52.3) | 308 (47.7) | |
| **Race** | | | | |
| Non-Hispanic white | 945 | 324 (33.53) | 621 (66.47) | <0.001 |
| Hispanic | 186 | 106 (53.34) | 80 (46.66) | |
| Non-Hispanic black | 194 | 94 (52.5) | 100 (47.5) | |
| **Age** | | | | |
| 20–39 | 34 | 10 (25.74) | 24 (74.26) | 0.001 |
| 40–59 | 233 | 80 (24.96) | 153 (75.04) | |
| 60 and over | 1116 | 454 (40.1) | 662 (59.9) | |
| Mean ± SE | 1383 | 67.40 (0.70) | 64.43 (0.64) | 0.002 |
| **Education** | | | | |
| High school graduate or less | 685 | 305 (40.49) | 380 (59.51) | 0.040 |
| Some college degree | 401 | 143 (35.17) | 258 (64.83) | |
| College graduate and above | 296 | 96 (29.17) | 200 (70.83) | |
| **Income** | | | | |
| Less 35000 | 593 | 263 (43.68) | 330 (56.32) | <0.001 |
| 35000–75000 | 407 | 153 (36.44) | 254 (63.56) | |
| Over 75000 | 253 | 71 (23.33) | 182 (76.67) | |
| **Birth Country** | | | | |
| US-born | 1216 | 459 (34.73) | 757 (65.27) | 0.030 |
| Mexico born | 126 | 66 (55.5) | 60 (44.5) | |
| Other | 40 | 19 (41.24) | 21 (58.76) | |
| **Health Insurance** | | | | |
| No | 67 | 35 (48.42) | 32 (51.58) | 0.078 |
| Yes | 1312 | 509 (35.16) | 803 (64.84) | |
| **Physical Activity** | | | | |
| No | 812 | 358 (40.92) | 454 (59.08) | 0.002 |
| Yes | 571 | 186 (30.03) | 385 (69.97) | |
| **Smoking** | | | | |
| Never | 618 | 275 (37.35) | 343 (62.65) | 0.595 |
| Current | 174 | 63 (36.22) | 111 (63.78) | |
| Former | 590 | 205 (33.67) | 385 (66.33) | |
| **Alcohol Use** | | | | |
| Never | 194 | 107 (54.35) | 87 (45.65) | <0.001 |
| Former | 319 | 141 (39.65) | 178 (60.35) | |
| Mild | 473 | 166 (34.69) | 307 (65.31) | |
| Heavy | 116 | 28 (21.39) | 88 (78.61) | |

[a]Unweighted sample size.

[b]Weighted percentage for categorical factors by applying NHANES sampling weights, or weighted mean and standard error (SE) for continuous variables (e.g., age).

[c]P-values are based on the Rao-Scott Chi-square tests for categorical variables or weighted linear regression for continuous variables. The P-value for comparing total ORC vs non-ORC proportions was obtained from Z-test.

[d]Estimate of the U.S. population, NHANES sampling weighted N, representing the results.

**Table 2. Multivariable logistic regression analysis of sociodemographic factors associated with obesity-related cancers in adults with metabolic syndrome, NHANES 2001–2018.**

| Characteristics | COR (95% CI)[a] | P-value[a] | AOR (95% CI)[b] | p-value[b] |
|---|---|---|---|---|
| **Time period** | | | | |
| Period 1 (ref) | - | - | - | - |
| Period 2 | 0.94 (0.65, 1.34) | 0.717 | 0.9 (0.59, 1.37) | 0.628 |
| Period 3 | 0.92 (0.63, 1.36) | 0.689 | 1.06 (0.67, 1.69) | 0.807 |
| **Gender** | | | | |
| Male (ref) | - | - | - | - |
| Female | 6.25 (4.43, 8.82) | <0.001 | 7.06 (4.85, 10.28) | <0.001 |
| **Race** | | | | |
| Non-Hispanic white (ref) | - | - | - | - |
| Hispanic | 2.27 (1.42, 3.61) | 0.001 | 2.86 (1.69, 4.83) | <0.001 |
| Non-Hispanic black | 2.19 (1.51, 3.18) | <0.001 | 2.65 (1.75, 4) | <0.001 |
| **Age** | | | | |
| 20–39 (ref) | - | - | - | - |
| 40–59 | 0.96 (0.38, 2.44) | 0.932 | 1.78 (0.61, 5.2) | 0.292 |
| 60 and over | 1.93 (0.78, 4.8) | 0.159 | 5.42 (1.9, 15.4) | 0.002 |
| **Education** | | | | |
| High school graduate or less (ref) | - | - | - | - |
| Some college degree | 0.8 (0.58, 1.09) | 0.158 | 0.87 (0.6, 1.28) | 0.485 |
| College graduate and above | 0.61 (0.40, 0.92) | 0.022 | 1.06 (0.65, 1.73) | 0.808 |
| **Income** | | | | |
| Over 75000 (ref) | - | - | - | - |
| Less 35000 | 2.55 (1.77, 3.67) | <0.001 | 1.25 (0.82, 1.89) | 0.302 |
| 35000–75000 | 1.88 (1.21, 2.93) | 0.006 | 1.31 (0.77, 2.23) | 0.328 |
| Missing | 2.15 (1.23, 3.75) | 0.008 | 1.14 (0.6, 2.18) | 0.688 |
| **Health Insurance** | | | | |
| No (ref) | - | - | - | - |
| Yes | 0.58 (0.31, 1.07) | 0.082 | 0.78 (0.37, 1.66) | 0.517 |
| **Physical Activity** | | | | |
| No (ref) | - | - | - | - |
| Yes | 0.62, (0.46, 0.84) | 0.002 | 0.8 (0.56, 1.13) | 0.203 |

[a]Crude odds ratio (COR) and p-value obtained from univariate logistic regression of an associated factor on ORC.

[b]Adjusted odds ratio (AOR) and p-value obtained from a multivariable logistic regression model of associated factors on ORC. The model was also adjusted for alcohol use, which was highly insignificant and not reported.

40.44% in 2001–2006 to 46.35% in 2013–2018, while the prevalence decreased for males from 13.73% in 2001–2006 to 8.17% in 2013–2018. This difference in ORC prevalence between males and females grew large over time. Among non-Hispanic black persons, age-standardized ORC prevalence was 62.56% in 2001–2006, declining to 42.16% in 2007–2012 and then substantially increasing to 54.72% in 2013–2018. On the other hand, among Hispanics, the most significant upward age-standardized ORC prevalence was observed, which increased from 23.23% in 2001–2006 to 63.13% in 2013–2018. A decreasing trend of ORC prevalence was observed for age groups 40–59 and 60+ years, while an increasing trend for the age group 20–39 years. Although for US-born people, the age-standardized ORC prevalence remained stable for the three time periods, the ORC prevalence changed dramatically for Mexican-born people in the first two periods, from 37.07% in 2001–2006 to 71.07% in 2007–2012, and then

**Table 3. Age-standardized and age-specific prevalence of ORC by associated factors.**

| Characteristics | T1: 2001–06 Wt % (SE)[a] | T2: 2007–12 Wt % (SE)[a] | T3: 2013–18 Wt % (SE)[a] | T3-T2 (95% CI)[b] | T3-T1 (95% CI)[b] | Beta[c] | P trend[c] |
|---|---|---|---|---|---|---|---|
| Overall | 30.55 (0.57) | 30.33 (0.38) | 30.65 (0.40) | 0.32 (-0.76, 1.40) | 0.1 (-1.26, 1.46) | 0.004 | 0.491 |
| **Gender** | | | | | | | |
| Male | 13.73 (0.50) | 9.01 (0.26) | 8.17 (0.2) | -0.84 (-1.48, -0.20) | -5.56 (-6.62, -4.50) | -0.278 | 0.137 |
| Female | 40.44 (0.80) | 45.24 (0.50) | 46.35 (0.62) | 1.11 (-0.45, 2.67) | 5.91 (3.98, 7.89) | 0.114 | 0.285 |
| **Race** | | | | | | | |
| Non-Hispanic white | 29.87 (0.45) | 25.15 (0.38) | 24.17 (0.41) | -0.98 (-2.08, 0.12) | -5.7 (-6.89, -4.51) | -0.134 | 0.205 |
| Non-Hispanic Black | 62.56 (1.27) | 42.16 (1.1) | 54.72 (0.78) | 12.6 (9.92, 15.2) | -7.84 (-10.8, -4.92) | -0.171 | 0.294 |
| Hispanic | 23.23 (0.64) | 50.03 (0.85) | 63.13 (0.94) | 13.10 (10.6, 15.6) | 39.9 (37.7, 42.1) | 0.828 | 0.004 |
| **Age group** | | | | | | | |
| 20–39[d] | 15.2 (1.74) | 20.79 (1.34) | 33.08 (1.30) | 12.3 (9.73, 14.9) | 17.9 (15.3, 20.5) | 0.541 | 0.283 |
| 40–59 | 25.37 (0.73) | 26.7 (0.61) | 23.07 (0.47) | -3.63 (-5.14, -2.12) | -2.3 (-4.00, -0.60) | -0.069 | 0.381 |
| 60+ | 42.86 (0.34) | 39.41 (0.31) | 39.16 (0.36) | -0.25 (-1.18, 0.68) | -3.7 (-4.67, -2.73) | -0.070 | 0.253 |
| **Education** | | | | | | | |
| High school grad or less | 40.38 (0.66) | 44.26 (0.51) | 30.86 (0.70) | -13.4 (-15.1, -11.7) | -9.52 (-11.4, -7.63) | -0.226 | 0.138 |
| Some college degree | 28.03 (0.81) | 21.33 (0.55) | 42.53 (0.79) | 21.2 (19.3, 23.1) | 14.5 (12.3, 16.7) | 0.422 | 0.063 |
| College grad or above | 23.28 (0.72) | 22.97 (0.54) | 17.93 (0.49) | -5.04 (-6.47, -3.61) | -5.35 (-7.06, -3.64) | -0.195 | 0.227 |
| **Income** | | | | | | | |
| Less 35000 | 30.8 (0.86) | 49.04 (0.37) | 43.15 (0.76) | -5.89 (-7.55, -4.23) | 12.4 (10.1, 14.6) | 0.234 | 0.173 |
| 35000–75000 | 36.63 (0.65) | 27.5 (0.69) | 33.42 (0.87) | 5.92 (3.74, 8.10) | -3.21 (-5.34, -1.08) | -0.034 | 0.448 |
| Over 75000 | 19.81 (0.68) | 14.16 (0.34) | 17.75 (0.45) | 3.59 (2.48, 4.70) | -2.06 (-3.66, -0.46) | 0.018 | 0.475 |
| **Country born** | | | | | | | |
| US-born | 29.67 (0.59) | 29.05 (0.40) | 28.66 (0.46) | -0.39 (-1.58, 0.80) | -1.01 (-2.48, 0.46) | | 0.447 |
| Mexico born | 37.07 (1.82) | 71.07 (0.71) | 56.07 (1.13) | -15.0 (-17.6, -12.4) | 19.0 (14.8, 23.2) | -0.024 | 0.427 |
| **Health Insurance** | | | | | | | |
| No | 57.59 (1.6) | 53.6 (0.78) | 42.97 (1.2) | -10.6 (-13.4, -7.82) | -14.6 (-18.5, -10.7) | -0.024 | 0.253 |
| Yes | 28.65 (0.61) | 27.65 (0.39) | 30.9 (0.41) | 3.25 (2.14, 4.36) | 2.25 (0.81, 3.69) | -0.078 | 0.351 |
| **Physical activity** | | | | | | | |
| No | 39.97 (0.80) | 33.99 (0.59) | 32.71 (0.65) | -1.28 (-3.00, 0.44) | -7.26 (-9.28, -5.24) | -0.134 | 0.284 |
| Yes | 26.97 (0.57) | 22.72 (0.35) | 27.42 (0.52) | 4.70 (3.47, 5.93) | 0.45 (-1.06, 1.96) | 0.020 | 0.459 |
| **Smoking** | | | | | | | |
| Never | 31.79 (1.02) | 25.73 (0.43) | 28 (0.62) | 2.27 (0.79, 3.75) | -3.79 (-6.13, -1.45) | -0.066 | 0.405 |
| Current | 47.47 (0.64) | 42.35 (0.89) | 26.38 (0.81) | -15.9 (-18.3, -13.6) | -21.1 (-23.1, -19.1) | -0.474 | 0.032 |
| Former | 22.14 (0.62) | 27.12 (0.55) | 37.82 (0.71) | 10.7 (8.94, 12.5) | 15.7 (13.8, 17.5) | 0.395 | 0.051 |
| **Alcohol use** | | | | | | | |
| Never | 59.61 (1.31) | 68.08 (0.76) | 38.97 (1.26) | -29.1 (-32.0, -26.2) | -20.6 (-24.2, -17.1) | -0.427 | 0.146 |
| Former | 39.48 (1.16) | 35.84 (0.79) | 29.78 (0.62) | -6.06 (-8.03, -4.09) | -9.7 (-12.3, -7.12) | -0.219 | 0.217 |
| Mild | 29.47 (0.66) | 27.7 (0.76) | 48.87 (1.0) | 21.2 (18.7, 23.7) | 19.4 (17.1, 21.8) | 0.534 | 0.042 |
| Heavy | 19.99 (0.99) | 20.73 (0.72) | 15.39 (0.56) | -5.34 (-7.13, -5.16) | -4.6 (-6.83, -2.37) | -0.180 | 0.310 |

[a]Weighted prevalence age-standardized by U.S. 2000 population. SE implies the standard error of the estimated prevalence.

[b]Percent difference in prevalence between the two periods. 95% confidence intervals (CI) were obtained using the Z test. The percent difference is significant if the CI includes 0 (p<0.05).

[c]Beta is obtained from univariate logistic regression of ORC on the continuous variable period(e.g., 1,2,3). It implies the rate of change in log odds from one time period to the following, implying an overall increase (+ve sign) or decrease (-ve sign) in the prevalence throughout the total study period. The p-value indicates the significance of a linear temporal trend in the prevalence.

[d]ORC prevalence for the age group 20–39 in period-1 may not be stable due to a small sample size (<10).

substantially decreased to 56.07% in 2013–2018. A decreasing trend in ORC prevalence was seen among higher or lower education groups, however, participants with some college degrees had a significantly increasing trend in ORC prevalence, from 28% in 2001–2006 to 42.53% in 2013–2018. Never and current smokers had a decreasing trend while former smokers had an increasing trend in ORC prevalence, from 22.14% in 2001–2006 to 37.82% in 2013–2018. Furthermore, the ORC prevalence significantly increased for mild alcohol drinkers from 29.47% in 2001–2006 to 48.87% in 2013–2018, however, within the first two periods the trend was stable.

### Prevalence and trends of individual ORCs

During the total period 2001–2018, the U.S. adults with MetS had only 6 specific ORCs out of 13 defined ORCs (Fig 2). These 6 ORCs were thyroid (10.95%), breast (10%), uterine (9.18%), colorectal (7.86%), ovarian (5.74%), and stomach (0.80%). The age-standardized prevalence of the top 5 ORCs in three periods was obtained (Fig 3). It showed a decreasing trend for breast cancers among adults with MetS. However, increasing trends were seen for colorectal, ovarian, thyroid and uterine cancers from period-1 to period-3.

## Discussion

About one-third of U.S. adults are currently affected by MetS, with a higher prevalence among older individuals, women, and those who are overweight [32, 33]. Obesity is a key factor in MetS and is linked to oxidative stress, which plays a role in several illnesses, including

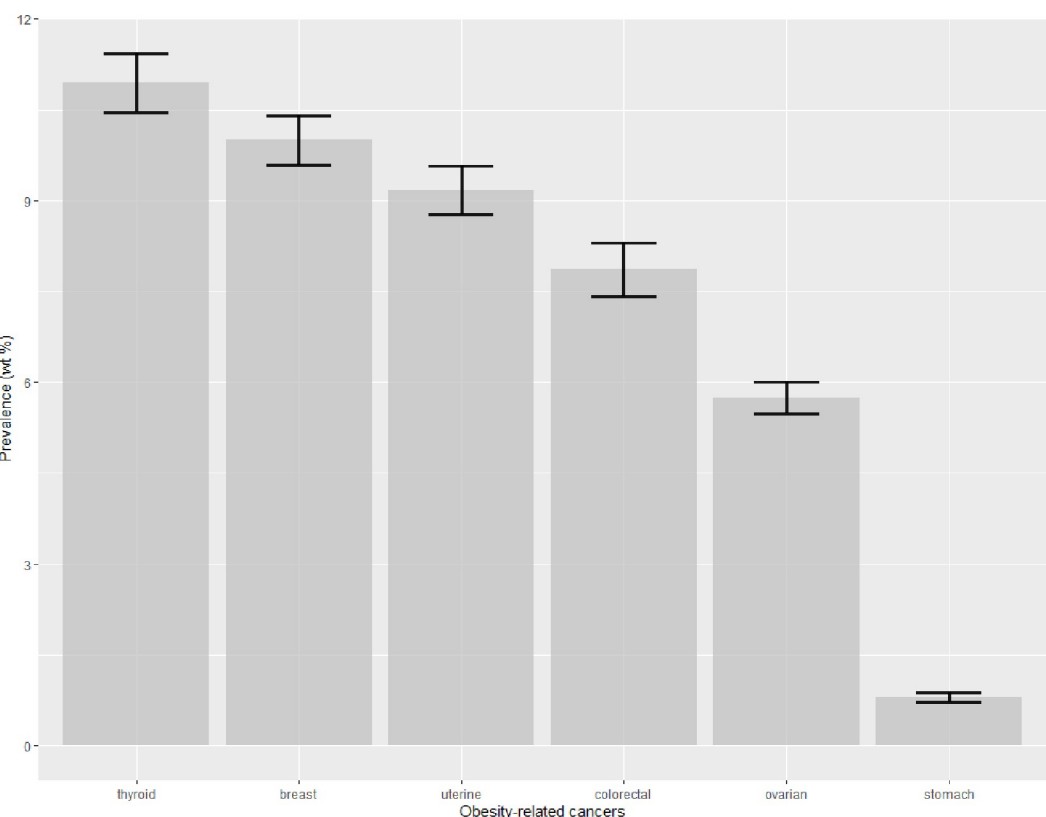

**Fig 2. Prevalence of obesity-related cancers among US adults with MetS, NHANES 2001–2018; error bars are "Prevalence ±SE".**

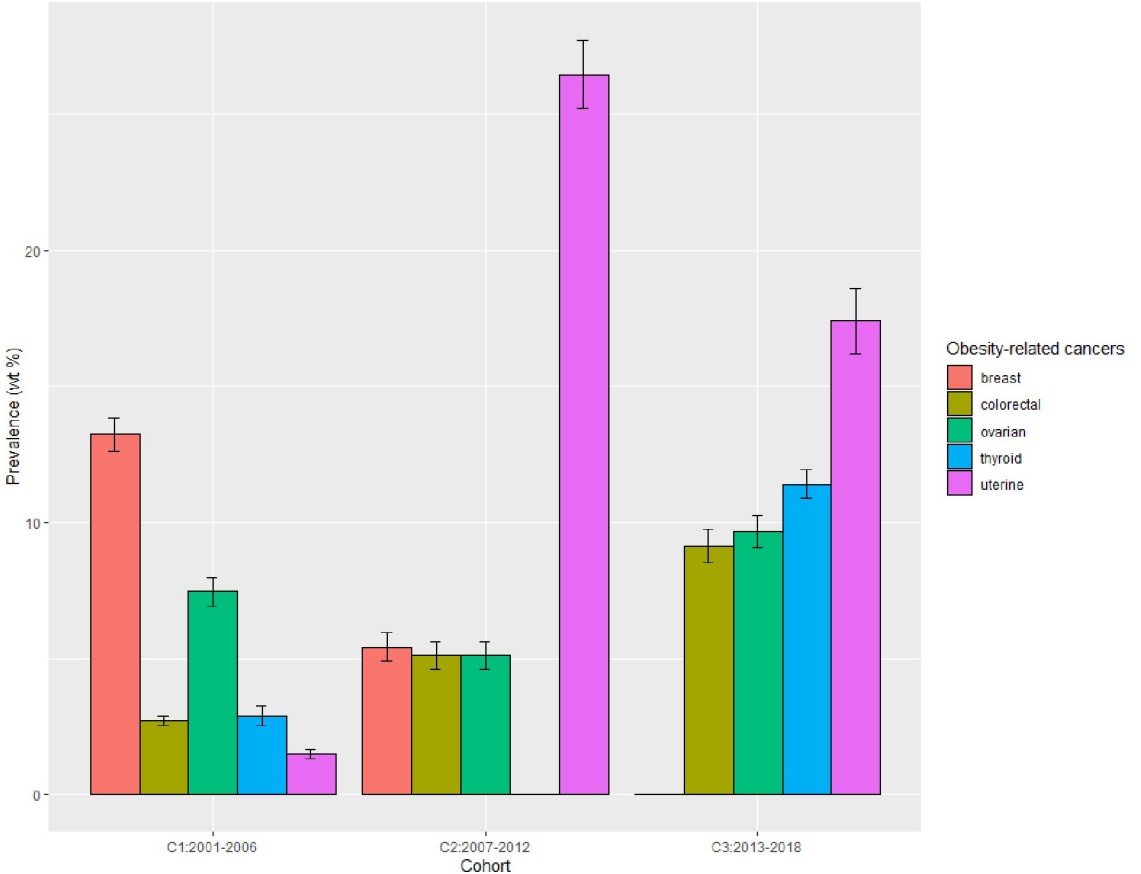

**Fig 3. Age-standardized prevalence of top 5 ORCs among U.S. adults with MetS in each time period; error bars are "Prevalence ± SE".**

cardiovascular disease, diabetes, and cancer [34, 35]. The present study aimed to identify groups with MetS who are at higher risk for developing ORC and temporal trends of ORCs among U.S. adults with MetS during the period of 2001–2018.

According to findings, the occurrence of ORC was more common in non-Hispanic Black and Hispanic participants with MetS, with a prevalence of over 50% for each group, as compared to non-Hispanic Whites with MetS. A higher prevalence was also observed among females with MetS compared to males. Even after controlling for various sociodemographic and lifestyle characteristics, this greater occurrence of ORC persisted as a significant finding. More specifically, the results from multivariable regression showed that females had a greater odds of developing ORC compared to males. This finding was consistent with several previous studies [6, 10]. Additionally, we observed that Hispanic and non-Hispanic black individuals displayed elevated odds in comparison to non-Hispanic whites from multivariable logistic regression. Similarly, Monestime et al. [6] found higher odds of ORC among Hispanic and non-Hispanic black individuals with MetS compared to non-Hispanic individuals with MetS, although they did not control behavioral factors, such as physical activity, and alcohol use. On the other hand, Steele et al. [4] analyzed U.S. cancer statistics data and found higher rates of ORCs among non-Hispanic Blacks and Whites compared to other ethnic groups. However, their study focused on the U.S. general population irrespective of MetS status. Steele et al. also revealed evidence that the frequency of overweight and ORC appearing together was greater in individuals of age 50 years or over, compared to those who were younger. Our findings also

supported these findings and found individuals aged 60 years and above exhibited significantly higher odds of developing ORC compared to those aged 20–39 years.

The individuals from these highly vulnerable groups may be more likely to consume a high-fat, high-calorie diet, which can contribute to obesity and metabolic syndrome. They may also be less likely to engage in regular physical activity, which can increase the risk of both obesity and cancer [36, 37].

The findings from temporal trend analysis indicate that the overall age-standardized prevalence of ORC remained stable across three time periods. However, there were some notable trends within subgroups. Specifically, there was a gradual increase in age-standardized ORC prevalence among females, while males exhibited a decrease over time. The difference in ORC prevalence between males and females also grew larger over time. Among non-Hispanic black individuals, ORC prevalence decreased from 2001–2012 but then increased substantially during 2013–2018. In contrast, among Hispanics, there was a significant upward trend in ORC prevalence. Interestingly, a decreasing trend in ORC prevalence was observed among both age groups 40–59 and 60+ years, while an increasing trend was observed in the age group 20–39 years. A study found that there was a significant increase in the incidence of several ORCs (multiple myeloma, colorectal, uterine, gallbladder, kidney, and pancreatic cancer) in young adults aged 25–49 years, indicating that the trend of increasing cancer incidence in young adults is accelerating over time [38]. A recent study by Contngco et al. [39] revealed that cancers linked to obesity are becoming more common among younger women, regardless of their race and ethnicity. The increasing burden of ORC in young adults can be attributed to conditions that often accompany excess body weight such as diabetes, gallstones, inflammatory bowel disease and poor diet.

According to the study's results, six out of the thirteen individual ORCs were prevalent among U.S. adults with MetS. The most common among these were thyroid, breast, uterine, colorectal and ovarian cancers (Fig 2). The incidence of breast cancer decreased over time among adults with MetS, but there were increasing trends for colorectal, ovarian, thyroid, and uterine cancers between 2001–2006 and 2013–2018 (Fig 3). An earlier study also found that the incidence in the U.S. population for breast cancer decreased while that for thyroid, ovarian, kidney, and renal pelvis cancers increased across birth cohorts from 1915 to 1985 [38]. A study that analyzed cancer registry data revealed a sharp 6.7% decline in the age-adjusted incidence rate of breast cancer among U.S. women in 2003, compared to 2002, which was mostly due to a sharp drop in hormone-replacement therapy (estrogen + progesterone) use [40, 41]. Additionally, the annual age-adjusted incidence rate decreased by 8.6% between 2001 and 2004 [40]. Based on recent data, breast cancer incidence rates increased by 0.5% annually from 2010 to 2019, while the death rate declined by 43% during the period of 1989 to 2020 [3]. Despite the fact that MetS is a recognized risk factor for breast cancer [42], the current study revealed a decreasing trend in breast cancer among U.S. adults with comorbid MetS and cancer. It is an interesting finding and requires further research to understand the relevant biological mechanisms or other related factors that potentially contributed to the decline in age-adjusted prevalence of breast cancer over time in this group of individuals.

Another study reported a decline in the incidence rate of colorectal cancer in the U.S. general population from 2005–2014 [4]. However, we observed the opposite trend for adults with MetS, which may be attributed to less likelihood of obtaining colorectal cancer screening [43] and difficulties in diagnosis [8]. Several other studies also linked metabolic syndrome to colorectal cancer [44, 45]. The current study's finding of the increasing trend in individual ORCs, excluding breast cancer, is also consistent with the rise in obesity prevalence in the U.S. during 2007–2014 [46]. Encouraging physical activity and promoting healthy eating habits within communities could help individuals maintain a healthy weight more easily.

The study utilized a large and nationally representative sample, and thus results can be generalized to the U.S. adults with MetS. This is the first study of its kind that examines temporal trends in the prevalence of ORC and identifies potential associated factors controlling for important behavioral characteristics. This study's findings might be useful for planning on preventing and controlling Mets, as well as emphasizing regular screening to further reduce the incidence of colorectal, ovarian, thyroid, and uterine cancers.

The limitation of this study is that it used cross-sectional data, which means it cannot establish the causality of the observed associations. Additionally, the study did not examine the impact of changes in dietary habits or other factors associated with quality of life on the occurrence of ORC.

## Conclusions

This study offers valuable insights into the prevalence and temporal trends of ORC among U. S. adults with MetS. The higher prevalence of ORC among females, Hispanics, and non-Hispanic blacks suggests that targeted interventions may be required for these groups to prevent ORC. Age was a significant risk factor for ORC, possibly due to age-related changes that predispose individuals to cancer. The stable overall prevalence of ORC over time, with increases in certain individual ORCs (thyroid, uterine, colorectal, and ovarian), indicates the need for more effective interventions to reduce the burden of ORCs among U.S. adults with MetS. Overall, this study underscores the importance of addressing MetS and lifestyle factors to prevent ORCs among U.S. adults with MetS.

## Supporting information

**S1 Checklist. STROBE statement—checklist of items that should be included in reports of observational studies.**
(DOCX)

## Author Contributions

**Conceptualization:** Harun Mazumder.

**Formal analysis:** Harun Mazumder, Maidul Husain, Md. Faruk Hossain.

**Methodology:** Harun Mazumder.

**Writing – original draft:** Harun Mazumder, Maidul Husain, Md. Faruk Hossain, Sultan Mahmud.

**Writing – review & editing:** Harun Mazumder, Maidul Husain, Md. Faruk Hossain, Sultan Mahmud.

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
