## [Decision Letter · Decision Letter 0]

28 Jul 2023

PONE-D-23-12854Prevalence, trend and risk factors of obesity-related cancers among U.S. adults with metabolic syndrome: Evidence from the National Health and Nutrition Examination Survey 2001-2018PLOS ONE

Dear Dr. Mahmud,

Thank you for submitting your manuscript to PLOS ONE. After careful consideration, we feel that it has merit but does not fully meet PLOS ONE’s publication criteria as it currently stands. Therefore, we invite you to submit a revised version of the manuscript that addresses the points raised during the review process.

We look forward to receiving your revised manuscript.

Kind regards,

Meisam Akhlaghdoust, M.D., M.P.H.

Academic Editor

PLOS ONE

Reviewers' comments:

Reviewer's Responses to Questions

**Comments to the Author**

1. Is the manuscript technically sound, and do the data support the conclusions?

Reviewer #1: Partly

Reviewer #2: Partly

Reviewer #3: Partly

2. Has the statistical analysis been performed appropriately and rigorously? 

Reviewer #1: No

Reviewer #2: No

Reviewer #3: Yes

3. Have the authors made all data underlying the findings in their manuscript fully available?

Reviewer #1: Yes

Reviewer #2: Yes

Reviewer #3: Yes

4. Is the manuscript presented in an intelligible fashion and written in standard English?

Reviewer #1: No

Reviewer #2: Yes

Reviewer #3: Yes

5. Review Comments to the Author

Reviewer #1: Title: Prevalence, trend and risk factors of obesity-related cancers among U.S. adults with metabolic syndrome: Evidence from the National Health and Nutrition Examination Survey 2001-2018

This study is about Prevalence, trend and risk factors of obesity-related cancers among U.S. adults with metabolic syndrome. The importance of this study is vague despite a similar study entitled ““Prevalence and predictors of obesity-related cancers among racial/ethnic groups with metabolic syndrome”, which was published in PLOS one journal”. By the way, I reviewed the paper and there are some comments as follows:

Abstract:

- Introduction: Please use related factors instead of predictors.

-in the results section, please identify the level of confidence for CI, i.e. (OR=7.1, 95% CI=4.9-10.3). “However, an increasing trend was seen for thyroid, uterine, colorectal and ovarian cancers while decreasing trend for breast cancer” . P value?

Introduction:

- The necessity and importance of this study is not clear. Although, you cited to a similar paper entitled “Prevalence and predictors of obesity-related cancers among racial/ethnic groups with metabolic syndrome”, which was published in PLOS one journal. But, you did not provide information about the difference between this study and yours. I know that the year of evaluation is different, and concentrated on ethnic groups. But, you could not justify the reason of conducting, which is necessary to notice that.

- “. This study aims to determine the prevalence and predictors of ORC based on recent data, and further evaluate the temporal trends in ORC 74 among U.S. adults with comorbid conditions MetS and cancer.” Please use past tense (e.g. This study aimed to …)

- This study is a cross-sectional study, which is differ from prediction model studies. Please use related factors instead of predictors. Because, the aim of this study is not to draw a prediction model and evaluate its performance. Thus, please revise it in the whole manuscript.

Methods:

- It is better to write this section according to STROBE guideline.

- There is no information on how to do a univariate logistic regression.

Results:

- Please report mean (SD) instead of mean (SE) in the table 1.

- How about univariate logistic regression analysis?

Discussion:

- Line 235-236: “Additionally, we observed that Hispanic and non-Hispanic black 236 individuals displayed elevated likelihood in comparison to non-Hispanic white”. Likelihood is not an appropriate term to use. You provide odds ratio measure. It is better to use odds instead of likelihood. If the out come prevalence is less than 5, you can use risk interchangeably. Please revise it in the whole document.

- Line 235- 239: “Additionally, we observed that Hispanic and non-Hispanic black individuals displayed elevated likelihood in comparison to non-Hispanic whites. Similarly, Monestime et al. (6) found higher odds of ORC among Hispanic and non-Hispanic black individuals with MetS compared to non-Hispanic individuals with MetS, although they did not control behavioral factors, physical activity, and alcohol use”. Adjusting for other confounders can not be the main reason to conduct another study.

Study limitation:

- The authors focused on the study strength, which can be mentioned while discussing. But, I think the major limitation is that investigator depended on the existing variables and such variables such as stress, quality of life and other factors was not considered. This section is very important, because it helps author researcher to design new studies.

Conclusion:

- In conclusion section, pleas not use reference. In this section, the authors should provide a conclusion on their own words.

Reviewer #2: The authors selected a subset of nhanes that may not adequately reflect the sampling weights used and would not allow an estimate of a population prevalence as claimed by the authors. For further concerns please, consult the attached file with details.

Reviewer #3: The document is quite interesting and addresses a subject that is increasingly being reviewed in the literature, that of ORCs and metabolic syndrome.

The document is methodologically and statistically well approached, however it leaves a bitter taste in my mouth that the findings are not fully discussed, I would have liked to see more interpretations and explanatory positions from the authors, I feel that they end up cutting it. I would like to point out, as an example, figure 3 in which the authors mention that breast cancer is one of those that present a decreasing trend over time, which can be seen in the figure, but it should have called attention that if it were the highest in the initial period, it declines until reaching almost zero in the last period. This must have caught the attention of the authors and I expected further discussion and a position from the authors on the matter.

6. PLOS authors have the option to publish the peer review history of their article (what does this mean?). If published, this will include your full peer review and any attached files.

Reviewer #1: No

Reviewer #2: No

Reviewer #3: **Yes: **Héctor Arreola-Ornelas

---

## [Author Response · Author response to Decision Letter 0]

16 Aug 2023

Response to Reviewer #1: 

Reviewer point #1: Abstract: Introduction: “Please use related factors instead of predictors.”

Author response #1:

We have replaced “predictors” with “associated factors”.

Reviewer point #2: Abstract: “in the results section, please identify the level of confidence for CI, i.e. (OR=7.1, 95% CI=4.9-10.3). “However, an increasing trend was seen for thyroid, uterine, colorectal and ovarian cancers while decreasing trend for breast cancer” . P value?”

Author response #2:

We agree that it is important to mention the level of confidence in CI and thus added in the abstract.

We did not report a p-value for the increase of thyroid, uterine, colorectal, and ovarian cancers. Because this conclusion on the increasing trend is evident from the time period-wise multiple bar charts with error bars with SE (Figure 3). Particularly, the highly non-overlapping SE error bars are implying evidence for statistical significance from one time period to another for a given cancer.

Reviewer point #3: Introduction: “The necessity and importance of this study is not clear. Although, you cited to a similar paper entitled “Prevalence and predictors of obesity-related cancers among racial/ethnic groups with metabolic syndrome”, which was published in PLOS one journal. But, you did not provide information about the difference between this study and yours. I know that the year of evaluation is different, and concentrated on ethnic groups. But, you could not justify the reason of conducting, which is necessary to notice that.”

Author response #3:

Thank you for your comment. This study clearly mentioned the gap in the current literature in lines 68-70 (However, less is known about the temporal trend in the prevalence of ORC among adults with MetS, though MetS is steadily increasing among U.S. adults). From lines 71 to 78, we also mentioned the specific aim of this study. In summary, we clearly indicated that our study is incorporating recent data to give a current and comprehensive picture of prevalence and predictors, which thereby implies no other study exists based on the publicly available most recent NHANES data in this ORC-MetS context. 

We agree that we cited a similar paper that includes data up to 2014 and focused on racial/ethnic groups, which is a point of discussion in our manuscript. In our discussion, we mentioned that their multivariable model did not adjust for two important variables in ORC and MetS (Physical activity and alcohol use) particularly when we discussed that our finding is supported by their finding for two racial groups (though our model shows slightly higher OR). We did not conduct our study only to show this. Our study clearly mentioned two other aims involving the trend in the prevalence of overall ORC and particular ORCs in the introduction section.

Reviewer point #4: Introduction: “. This study aims to determine the prevalence and predictors of ORC based on recent data, and further evaluate the temporal trends in ORC 74 among U.S. adults with comorbid conditions MetS and cancer.” Please use past tense (e.g. This study aimed to …)

Author response #4:

We agree with the reviewer on this comment. And revised accordingly.

Reviewer point #5: Introduction: “This study is a cross-sectional study, which is differ from prediction model studies. Please use related factors instead of predictors. Because, the aim of this study is not to draw a prediction model and evaluate its performance. Thus, please revise it in the whole manuscript.”

Author response #5:

We agree with the reviewer that our purpose is not the prediction. Thus, we have used “associated factors” instead of “predictors”. We have incorporated this adjustment throughout the manuscript as per your recommendation. Thank you for your valuable feedback.

Reviewer point #6: Method: “It is better to write this section according to STROBE guideline.”

Author response # 6:

We have already prepared the method section according to STROBE guidelines for the submitted manuscript. We provided information on applicable STROBE items in our study such as study design and participants, variables, data source and analytical sample selection, study size, statistical methods, missing data, ethics, and subgroup analyses. We provided a diagram showing detailed steps involving the selection of the study participants/analytical sample. The effort to reduce bias was part of the NHANES survey design and data collection, thus not needed in our study. In addition, we have also submitted the STROBE checklist along with the manuscript.

Reviewer point #7: Method: “There is no information on how to do a univariate logistic regression.”

Author response #7:

Thank for your comment. We have added details on univariate logistic regression in method section. In addition, we have added the result from this regression in the result section. 

Reviewer point #8: Results: “Please report mean (SD) instead of mean (SE) in the table 1.”

Author response #8:

We respectfully disagree with this comment. According to NHANES guidelines and tutorials, reporting SE is important to understand the stability and admissibility of the estimated prevalence values. According to the NHANES analytical guidelines, if standard errors are more than 30% of the prevalence estimates, then the results may not be stable. [Comment from NHANES tutorial: The NHANES guidelines recommended a relative standard error of at most 30%.] (https://wwwn.cdc.gov/nchs/nhanes/tutorials/ReliabilityOfEstimates.aspx ). Therefore, it is necessary to SE instead of SD.

Furthermore, SE represents the uncertainty and is important to get an idea of the confidence interval of the estimates. Therefore, we think SE is more meaningful and necessary to be reported instead of SD. For NHANES data analysis, it is common to report SE instead of SD in the literature.

Reviewer point #9: Result: “How about univariate logistic regression analysis?”

Author response #9:

Thank you for your question. Now, we have added the univariate logistic regression analysis. 

Reviewer point #10: Discussion: “Line 235-236: “Additionally, we observed that Hispanic and non-Hispanic black 236 individuals displayed elevated likelihood in comparison to non-Hispanic white”. Likelihood is not an appropriate term to use. You provide odds ratio measure. It is better to use odds instead of likelihood. If the outcome prevalence is less than 5, you can use risk interchangeably. Please revise it in the whole document.”

Author response #10:

The word likelihood is replaced with odds in the whole manuscript for describing our study findings. 

Reviewer point #11: Discussion: “Line 235- 239: “Additionally, we observed that Hispanic and non-Hispanic black individuals displayed elevated likelihood in comparison to non-Hispanic whites. Similarly, Monestime et al. (6) found higher odds of ORC among Hispanic and non-Hispanic black individuals with MetS compared to non-Hispanic individuals with MetS, although they did not control behavioral factors, physical activity, and alcohol use”. Adjusting for other confounders can not be the main reason to conduct another study.”

Author response #11:

Thank you for your comment. I would like to clarify that we did not assert conducting another study merely by adjusting two additional important factors. We mentioned this study to show that our finding is supported by a study with a similar findings for racial/ethnic groups. However, our main objective is to give a current and comprehensive picture of prevalence and associated factors of ORC among the study population. 

[We mentioned: Similarly, Monestime et al. (6) found higher odds of ORC among Hispanic and non-Hispanic black individuals with MetS compared to non-Hispanic individuals with MetS, although they did not control behavioral factors, such as physical activity, and alcohol use.]

Reviewer point #12: Study limitation: “The authors focused on the study strength, which can be mentioned while discussing. But, I think the major limitation is that investigator depended on the existing variables and such variables such as stress, quality of life and other factors was not considered. This section is very important, because it helps author researcher to design new studies.”

Author response #12:

We thank and agreed with the reviewer on this point that we relied on the existing variables and variables such as stress, quality of life and others were not considered in this study. Our study focused on common risk factors such as demographic and behavioral factors. Therefore, controlling for stress and quality of life is beyond the scope of this study.

We have revised the manuscript by adding this as the limitation at the end of the discussion section.

Reviewer point #13: Conclusion: “In conclusion section, please not use reference. In this section, the authors should provide a conclusion on their own words.”

Author response #13:

We agree that the conclusion should not have a reference mentioned. We removed that and revised the conclusion in the updated manuscript.

Response to Reviewer #2: 

Reviewer point #1: “The authors selected a subset of nhanes that may not adequately reflect the sampling weights used and would not allow an estimate of a population prevalence as claimed by the authors. For further concerns please, consult the attached file with details.”

The remarks from the reviewer in the attached file for the authors:

Remark 1:

“There are several issues with the approach chosen by the authors. First, only subjects with a cancer diagnosis were selected. If this subset differs substantially from the subset of the population that is free of cancer it is possible to miss a risk factor that is specifically related to ORC but not a risk factor for cancer in general. The design chosen by the authors allows only the identification of factors relative to a ORC vs non-ORC cancer diagnosis. It is not clear that the weights calculated in the NHANES survey adjust for the population diagnosed with cancer. If that is the case, this needs to be stated explicitly. If not, then the weights used do not project onto the US population but only onto the cancer population. Thus, it Is not possible to calculate a prevalence of ORC but only to determine the proportion of ORC’s among all the cancers.”

Author response #1:

Thanks for your detailed comment. We agree that our design allows the identification of factors related to ORC vs non-ORC cancer diagnosis among US adults with Metabolic Syndrome (MetS). For example, the focus of the study is to estimate the prevalence of ORC and associated risk factors of ORC (response variable: ORC=1 vs No-ORC=0) among individuals with MetS. Our main subpopulation is the individuals having MetS. NHANES calculates the sampling weights (e.g., fasting subsample) by focusing more on the multi-stage nature of the survey (e.g., secondary sampling unit is nested under the primary sampling unit), the distribution of age and race (better representation by oversampling), and of course the non-response in each phase of the survey/data collection (e.g., home interview, medical examination, fasting glucose collection, etc.), NOT specifically accounting for any particular disease. The sampling weights used in this study were directly related to the components of MetS i.e., non-response adjustment during obtaining fasting glucose level. The diagnosis of cancer did not have an impact on the sampling weight calculation. Therefore, focusing on the ORC vs non-ORC does not potentially prevent us from constructing a subpopulation of individuals with MetS that represents the US population of age 20 years or older.

We also want to point out that we first defined the survey design (using survey R package) on the full population, then we obtained a subset of this survey design according to our subpopulation of interest. In this way, the survey package adjusted the sampling weights according to the subpopulation (adults 20 years or older and having MetS) so that the estimates are representative of the US population in this group.

A paper focusing on risk factors of ORC vs non-ORC among MetS individuals also reported results that projected on the US population with MetS, not just the individuals with a cancer diagnosis. Our study also has the same study population (individuals with comorbid metabolic syndrome and cancer mentioned in their study population section). https://journals.plos.org/plosone/article?id=10.1371/journal.pone.0249188

Remark 2:

“The bivariate analyses, chi-squared tests chosen by the authors should only serve as a selection for relevant variables in the logistic regression model. Since the cancer cases are not a random subset of all surveyed subjects, it is not clear that the bivariate parameters have any interpretation as marginal parameters for the US population. For risk factor assessment, as the authors state, causal statements are not possible, therefore, any factor potentially related to the type of cancer should be stated as being associated with ORC or not. Furthermore, other variables such as age that are likely confounders, need to be adjusted for.”

Author response #2:

Our focus was to consider the behavioral and demographic factors. The bivariate analyses and chi-squared tests were useful in three ways: 1) one is the selection of important variables for the multivariable logistic regression model, 2) To estimate the prevalence of ORC within a subcategory of the population; for example, ORC prevalence among males with MetS. 3) To identify any significant differences in ORC prevalence between the subgroups; for example, whether the prevalence of ORC in males and females were different or not. Since this is a univariate type of analysis, the results were not adjusted for age in Table 1. We agree that age is a confounder and therefore, we adjusted for age in the risk factor assessment in Table 3. We also adjusted for age distribution across time periods to make them comparable and obtain a trend in prevalence of ORC presented in Table 2.

We agree with the remark that “For risk factor assessment, as the authors state, causal statements are not possible, therefore, any factor potentially related to the type of cancer should be stated as being associated with ORC or not”. In our writing on risk factor assessment, we did not indicate causation; we indicated association. To make it easier to follow we now used the word “associated factors” instead of “risk factors”. We have also revised the manuscript by replacing risk factors with associated factors or related factors.

Remark #3: 

“The term cohort for the three groups of years is inappropriate. A cohort is a group that is selected and then observed forward. This is not the case here. There are 18 separate samples, one for each year. It is questionable to assess trends in any meaningful way over time since risk due to such factors as age, year born etc are confounded with time trends. Also, it does not seem that the authors directly tested trends over time except to see if cohort as the groupings are named is a predictor of outcome.”

Author’s response:

Thank you for catching this up. We changed the term “cohort” to “time period” everywhere in the manuscript. 

We agree that the age distribution of the populations in each of the NHANES cycles and thus in each of the three periods are different. Therefore, we applied the age-standardization method based on the US 2000 population so that age distribution and thus the estimated prevalences are comparable across the time periods. The age-standardized estimates are also called the age-adjusted estimates. We provide more clarification in the statistical analysis section. 

Remark #4: 

“Also, it should be stated somewhere in the paper why the cancers chosen are called ORC. This is not obvious and obviously, not every breast or uterine cancer or any other cancer is due to obesity. Is it even known what proportion of these cancers are attributable in the population to obesity? Why was BMI or a similar measure not used as a covariate?”

Author’s response:

In the introduction section we mentioned: Obesity has been found to be linked to a higher risk for cancer in at least 13 anatomic sites (reference #: 2,4,6,7).

In the study variables section we provided references about defining ORC:

A diagnosis of brain, bladder, esophagus, kidney, endometrial, thyroid, ovarian, breast, liver, gallbladder, stomach, colorectal, or pancreatic cancer was required for the presence of an ORC to be established (reference #: 2,4,6,7).

Again, we agree that no cancers can be assigned 100% due to obesity. But for these 13 cancers, research shows obesity played a significantly greater risk. In the case of uterine cancers, it is well-known that a vast majority of uterine cancers are endometrial and obesity was found to be the strongest risk factor. We looked at several studies which defined these cancers as the ORC. Majority of the cancers occur in individuals of age over 40-50 years old. It is well known that after menopause, obesity significantly increases the risk of breast cancer.

We did not consider BMI as one predictor because it is correlated with the metabolic syndrome component variables.

Remark #5:

“Many of the cancers are age-related and it is not clear to what degree these ORC are due to age and to what degree due to obesity. If a person is not overweight and does not have metabolic syndrome, then whatever cancer they have, obesity is not the cause. If the purpose is to assess the degree to which obesity and metabolic syndrome contribute to cancer burden, it should bestudied in the population that includes non-cancer cases as well.”

Author’s response #5:

We somewhat agree with this remark from the reviewer. H owever, our specific focus was ORC relative to non-ORC among individuals with MetS. Maybe the reviewer’s this comment can be a direction for future research.

Some specific remarks/corrections:

Line 69: and reduced likelihood...screening and difficulties... the authors refer to 2 different

issues.

Line 72: ninimize the risk

Line 82: two-year cycle

Line 95/96: elevated triglycerides stated twice

Line 129: reference for direct method of standardization 

Line 132: what Z-test was used?

Line 133: was time included as year beginning with either 2001 or 1? Also, it should be stated

that a linear trend was assessed on the logit scale.

Line 170: omit THE and start the sentence with Multivariable....

Line 171: found to be..

Line 172: higher than in males

Line 195: stable for the three cohorts

Line 233 from multivariable logistic regression

Line 265: MetS with thyroid....cancers being the most...

Line 271: MetS, which maybe attributable to unless it is known to be the case.

Author’s response for the above remarks #:

We agree to all these corrections and remarks, and revised the manuscript accordingly, except for “Line 129: reference for direct method of standardization”. Because the reference for the direct method of standardization is already provided in the submitted manuscript. In line 132, two proportions Z-test was used and in line 133 the time included as year beginning with 1. We added a sentence stating linear trend was assessed on the logit scale.

Response to Reviewer #3: 

Reviewer point #1: “The document is quite interesting and addresses a subject that is increasingly being reviewed in the literature, that of ORCs and metabolic syndrome.

The document is methodologically and statistically well approached, however it leaves a bitter taste in my mouth that the findings are not fully discussed, I would have liked to see more interpretations and explanatory positions from the authors, I feel that they end up cutting it. I would like to point out, as an example, figure 3 in which the authors mention that breast cancer is one of those that present a decreasing trend over time, which can be seen in the figure, but it should have called attention that if it were the highest in the initial period, it declines until reaching almost zero in the last period. This must have caught the attention of the authors and I expected further discussion and a position from the authors on the matter.”

Author response #1:

Thank you for your valuable suggestion. More discussions have been added on this declining trend in the age-adjusted prevalence of breast cancer over time. The authors want to point out that, although some studies reported a decline in the incidence rate of breast cancer during the first time period (2001-2006), further research is warranted to better understand the underlying factors that potentially contributed to this decline among individuals with metabolic syndrome.

---

## [Editor Report · Decision Letter 1]

21 Aug 2023

Prevalence, trend and associated factors of obesity-related cancers among U.S. adults with metabolic syndrome: Evidence from the National Health and Nutrition Examination Survey 2001-2018

PONE-D-23-12854R1

Dear Dr. Mahmud,

We’re pleased to inform you that your manuscript has been judged scientifically suitable for publication and will be formally accepted for publication once it meets all outstanding technical requirements.

Kind regards,

Meisam Akhlaghdoust, M.D., M.P.H.

Academic Editor

PLOS ONE
---

## [Editor Report · Acceptance letter]

24 Aug 2023

PONE-D-23-12854R1 

Prevalence, trend and associated factors of obesity-related cancers among U.S. adults with metabolic syndrome: Evidence from the National Health and Nutrition Examination Survey 2001-2018 

Dear Dr. Mahmud:

I'm pleased to inform you that your manuscript has been deemed suitable for publication in PLOS ONE. Congratulations! Your manuscript is now with our production department. 

Kind regards, 

on behalf of

Dr. Meisam Akhlaghdoust 

Academic Editor

PLOS ONE